# Effect of intensity training block on anxiety state and performance in competitive swimmers

Hajer Aouani[1,2], Sofiene Amara[1,2], Faten Sahli[1,2], Tiago M. Barbosa[3,4], Nizar Souissi[1,5] and Roland van den Tillaar[6]

[1] Higher Institute of Sport and Physical Education of Ksar-Said, University of La Manouba, Tunis, Tunisia
[2] Research Unit (UR17JS01) Sports Performance, Health & Society, Higher Institute of Sport and Physical Education of Ksar Saîd, Universite de la Manouba, Tunis, Tunisia
[3] Research Center in Sport, Health and Human Development, Vila Real, Portugal
[4] Department of Sports Sciences, Instituto Politécnico de Bragança, Campus Sta., Bragança, Portugal
[5] Physical Activity, Sport, and Health, UR18JS01, National Observatory of Sport, Tunis, Tunisia
[6] Department of Sport Sciences and Physical Education, Nord University, Levanger, Norway

Corresponding authors
Sofiene Amara,
coachsofieneamara@gmail.com
Roland van den Tillaar,
roland.v.tillaar@nord.no

## ABSTRACT

**Background:** An increase in training intensity could create changes in psychological and physiological variables in competitive athletes. For this reason, it is very relevant to know how an intensive training block could influence psychological variables in competitive swimmers. This study examined the effect of an intensive training block (HIT) for 2 weeks on the anxiety state and swimming performance compared to standard aerobic training.

**Methods:** Twenty-two male competition swimmers were randomly assigned to two groups: HIT group ($n = 11$; age = $16.5 \pm 0.29$ years) and control group following the standard training program ($n = 11$; age = $16.1 \pm 0.33$ years). Psychological status variables (cognitive anxiety, somatic anxiety and self-confidence) and swimming performance (100-m front crawl) were measured pre-and post-test.

**Results:** A significant effect of time was found for all psychological variables and swimming performance ($F \geq 17.6$; $p < 0.001$; d $\geq 0.97$). Furthermore, a significant group × time interaction effect was found in cognitive ($F = 14.9$; $p < 0.001$; d = 0.62) and somatic anxiety ($F = 5.37$; $p = 0.031$; d = 0.55) were found. Only a significant group effect was found in somatic anxiety ($F = 27.1$; $p < 0.001$; d = 1.2). *Post hoc* comparison revealed that both groups increased their cognitive anxiety and swimming performance, and decreased their self-confidence from pre to post test. However, cognitive anxiety increase significantly more in the HIT group compared to the control group. Furthermore, only the HIT training group significantly increased somatic anxiety over time, while somatic anxiety did not change significantly over time in the control group.

**Conclusion:** Our findings indicated that a sudden increase in training intensity increased state anxiety more than standard training, but both conditions similarly enhanced swimming performance. Although the current level of psychological state is not affecting swimming performance negatively over this period, it should be regularly monitored by psychologists as it over a longer training period perhaps could have a negative influence on swimming performance.

# INTRODUCTION

To reach elite level in swimming, several abilities (*e.g.* physical, psychological) must be developed during a swimmer's career (*Amara et al., 2022*; *Chortane et al., 2022*; *Amara et al., 2023*). Mental development promotes physical and physiological adaptation in swimmers during training and competition (*Clemente-Suárez et al., 2021*). Coaches and swimmers develop their training strategies (*i.e.* handling the training load) by taking the days of the competitions into consideration (*i.e.* major competition, national competition) (*Clemente-Suárez et al., 2017*; *Arroyo-Toledo et al., 2013*). Increases or decreases of the training load (*e.g.* volume, intensity and frequency) could affect positively or negatively the emotional state of swimmers (*Chortane et al., 2022*). On the basis of this idea, it is necessary to gather more information on the psychological trajectories over intensive training periods in competitive swimmers.

Intensive training is a very common methodology in swimming, aiming to improve several physical qualities (*e.g.*, aerobic ability, anaerobic ability) (*Nugent et al., 2019*). High intensity training (HIT) is an intermittent training methodology characterized by very high intensity (over 80% of maximal heart rate) and short duration (<20 s) and very high energy capacity (*Nugent et al., 2019*; *Costill et al., 1988*). In the same context, Ultra Short Race Pace Training (USRPT) is a very common high intensity training method performed by competitive swimmers (*Nugent et al., 2017*; *Hawley et al., 1997*). This training method (USRPT) was classified as a derivative of HIT, based as repeated bouts of high-intensity training (does not exceed 20 s) from maximal lactate steady state to supramaximal training intensity, with recovery periods of low intensities or complete rest (*Nugent et al., 2017*; *Hawley et al., 1997*). Notwithstanding, *Costill et al. (1988)* showed that swimmers experienced local muscular fatigue and difficulty in finishing the workouts of a block of HIT (10-days) with an intensity set at 94% VO2 max. These authors (*Costill et al., 1988*) suggested that swimmers must have experience in intensive training and a certain level of physical fitness to manage successive intensive exercises and reduce the risk of musculoskeletal injury and fatigue. Moreover, monitoring the psychological state (*e.g.* cognitive anxiety, somatic anxiety, self-confidence) during HIT is paramount to detect the symptoms of overtraining and better adapting to the successive stimuli of intensive training (*Goss, 1994*). For this, the current study aims to provide some insight into how anxiety is affected by intense block training in competitive swimmers.

Anxiety is an emotional state expressed by feelings of nervousness and anxiety that are assessed by psychometric questionnaires (*e.g.* CSAI-2R, STAI-Y) (*Cox, Martens & Russell, 2003*; *Julian, 2011*). Anxiety, whether cognitive (*i.e.* Apprehensions and mental tensions) or somatic (*i.e.* physiological manifestations of anxiety), might be strongly influenced by an athlete's ability to meet psychological and physiological demands (*Chortane et al., 2022*). According to literature, the state of anxiety has been treated as a normal personality trait or as a pathology (*Cox, Martens & Russell, 2003*; *Laurencelle & André, 2011*). What we have

noticed is that previous research focuses on physiological, physical and biomechanical studies in swimmers; whereas, studies of psychological trajectories (*e.g.* cognitive anxiety, somatic anxiety and self-confidence) require further clarification, remaining yet elusive (*Lopes et al., 2021*; *Sammoud et al., 2021*). Nonetheless, some have studied the effect of training on the state of anxiety in swimmers (*Dalamitros et al., 2019*; *Vacher et al., 2017*). For example, a study by *Dalamitros et al. (2019)* evaluated the effect of lower limb strength exercise (five box jumps) on anxiety, swimming performance (50-m breaststroke) and heart rate in both expert and non-expert swimmers. These authors (*Dalamitros et al., 2019*) showed that potentiating activity performed during dry land training (*i.e.* five-box jump) was insufficient to induce desirable changes in the specific psychological and physiological variables tested. On the other hand, given the specificity of swimming, the physical and psychological responses after intensive aquatic training could be different from those after intensive training on dry land. For this, it is also important to study the effect of high-intensity aquatic training on psychological variables (*e.g.* anxiety, stress) and swimming performance in competitive swimmers.

A swimming HIT block might lead to physiological disturbances and strain (muscle fatigue, metabolic maladjustment) (*Costill et al., 1988*) that in turn can lead to changes in the psychological state of competitive swimmers. In the same context, *Clemente-Suárez et al. (2021)* had shown that swimmers' anaerobic performance correlated to psychological characteristics (*i.e.* low stress r: 0.526, and high fatigue r: −0.506). Based on these ideas, it is necessary to know more information regarding the effect of a block of high intensity training on the psychological state in swimmers, in order to help them to be in good condition psychological and physical during the pre-competition period or on the day of competition.

The aim was to study the effect of a block of HIT (2-week) on psychological state (*e.g.* cognitive anxiety, somatic anxiety and self-confidence) and swimming performance (100-m front crawl) in competitive swimmers. We hypothesized that intensive training increases anxiety among swimmers, while 2 weeks of training would not improve swimming performance due to the short time.

## MATERIALS AND METHODS

### Study design

A randomized controlled trial was designed to assess the effect of an intensive training microcycle (2-week) on the psychological state (somatic anxiety, cognitive anxiety and self-confidence) and swimming performance (100-m front crawl) in competitive swimmers. All independent variables were measured before and after the intervention period (Fig. 1). This study was conducted in a 25 m indoor pool with water and air temperatures of 27.1 and 25.9 °C, respectively, and 64% relative humidity during the swimming sports season (October to November).

### Participants

Twenty-two male competition swimmers were randomly assigned into two groups. High-intensity training group (HIT group: $n$ = 11, age = 16.5 ± 0.29 years; size: 177 ±

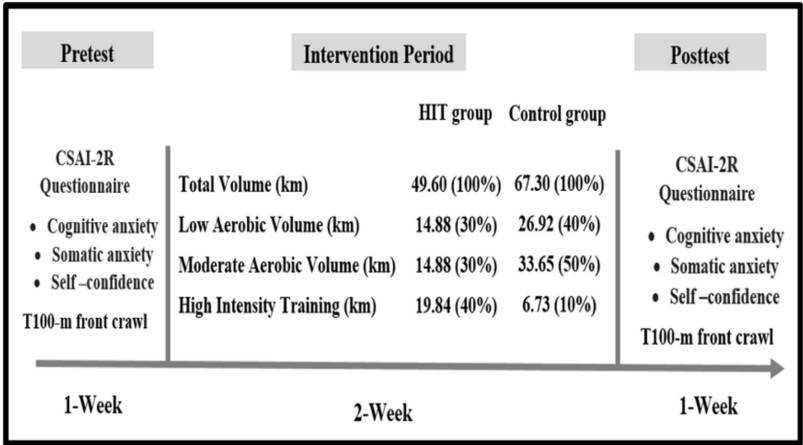

**Figure 1  Study design in the both groups.**   

8.82 cm; body mass = 74.5 ± 5.36 kg) and control group as part of a usual training ($n$ = 11, age = 16.1 ± 0.33 years; height: 176 ± 8.40 cm body mass 74.6 ± 5.11 kg). *A priori* power analysis (G * Power 3.1.9.3; Heinrich Heine Universität Düsseldorf, Düsseldorf, Germany) for ANOVA tests family (between- within interaction) yielded a total sample size of at least 20 swimmers to detect significant effects (actual power = 0.84; effect size = 0.35) assuming a power of 0.8 and alpha of 0.05. All swimmers were briefed on all the details of the design of the study and swam under the same coaches. The best and the worst performance time in the 100 m crawl was 55.22 and 58.70 s respectively. All competitive swimmers selected to participate in this pre-sent study had over six years of experience in training and competition at the national level, participated in the last Tunisian swimming championship and were ranked among the top thirty in the 100-front crawl test. Any swimmer with an injury requiring more than 15 days of rest or missing more than 10% of training sessions during the past 6 months was excluded from participating in this study. All participants and parents read and signed written informed consent. This study was approved by an institutional review committee of the Higher Institute of Sport and Physical Education of Ksar Said, Manouba University, Tunisia (Sports Performance Research Unit, Health and Society, UR17JS01, protocol code 2022-167), and was established according to the latest Helsinki declaration.

## Swimming training

The two swimming training protocols (HIT and standard training) were composed of three training categories (*Chortane et al., 2022*; *Amara et al., 2023*; *Costill et al., 1988*):

(i) Low aerobic training (LAT): warm-up, technical exercises *i.e.*, 800, 4 × 400 m; at 50% to 60% of maximum heart rate (HRmax).

(ii) Moderate aerobic training (MAT): aerobic training *i.e.*, 6 × 200 m, 10 × 100 m; at 60% to 80% of HRmax.

(iii) High-intensity training (HIT): Ultra Short Race Pace Training ; *i.e.*, 8 × 25 m, 2 × 6 (15 m) at 80% to 95% of HRmax.

The HIT group was invited to perform an intensive low-volume training (total distance = 46.60 km): 60% aerobic training, 40% HIT. In return, the control group followed the standard training program characterized by a higher training volume (total distance = 67.30 km): 90% aerobic training, 10% HIT (Table 1, Fig. 1). Rating of perceived exertion (RPE score, scale 1 to 10) was assessed after 30 min of each session during the last 2 weeks of standard training before the IP and during the 2 weeks of the intervention period (Borg, 1982). Rating of perceived exertion load (RPE load) by multiplying the swimmer's perceived effort assessment (RPE score) by the training volume (duration) of the session (Table 1) (Foster et al., 2001; Wallace, Slattery & Coutts, 2009).

## Measures of psychological status variables

The CSAI-2R questionnaire (French version) was implemented 1 h before the 100 m front crawl test to assess the psychological status of swimmers (Chortane et al., 2022; Cox, Martens & Russell, 2003). All swimmers were asked to complete the CSAI-2R questionnaire, which consists of 17 items and is divided into three blocks (Martinent et al., 2010): (i) block 1 of 5 items with an intensity between 5 and 20 to assess cognitive anxiety. (ii) block 2 of 7 items ranging in intensity from 7 to 28 is designed to assess somatic anxiety. (iii) Block 3 of 5 items with an intensity set between 5 and 20 to assess self-confidence. The questionnaire response was evaluated on a four-point scale: (1) "Not at all", (2) "A little", (3) "Moderately" and (4) "Very well." The score for each block was calculated by adding up, dividing by the number of items, and multiplying by 10. Re-liability (ICC) between pre-test and post-test of all psychological variables ranged from 0.83 to 0.88.

## Swimming performance test

A standard warm-up was performed by all swimmers and consisted of an aerobic (800-m) and progressive sprint ($8 \times 25$ m) sets. The 100 m front crawl time-trial from a block start was measured by two expert timekeepers using stopwatch (SEIKO S120-4030; Seiko, Tokyo, Japan). The 100-m test was conducted in the morning (Amara et al., 2021). The intraclass correlation coefficient (ICC) for the pre-test and post-test reliability was 0.91.

## Statistical analysis

Data sets are processed using SPSS version 26 for Windows (SPSS Inc., Chicago, IL, USA). The sphericity and normality of the datasets were verified using the Mauchly and Shapiro-wilk's tests, respectively. Reliability measurements between pretest and posttest were calculated using the Intraclass Correlation Coefficient (ICC) test (Weir, 2015). To determine the effect of an intensive training block (HIT) on the anxiety state and swimming performance a 2 (group: HIT vs. control) × 2 (test time: pre- post, repeated measurement) ANOVA tests were performed on all variables. Post hoc testing using Holm–Bonferroni probability adjustment was used to locate significant differences. Post hoc The effect size (ES) was determined by converting a partial squared state to Cohen's d

**Table 1 Mean load training in both groups during the 2 weeks of training.**

| Groups | Weeks | Mean distance (m) | Mean duration (min) | Mean RPE (a.u) | Mean RPE Load (a.u) |
|---|---|---|---|---|---|
| HIT group | Week 1 | 4,083.33 ± 116.95 | 78.83 ± 1.94 | 7.33 ± 0.61 | 578.42 ± 53.86 |
| | Week 2 | 4,183.30 ± 116.91 | 79.83 ± 3.76 | 7.92 ± 1.20 | 635.58 ± 120.98 |
| Control group | Week 1 | 5,550 ± 216.80 | 101.00 ± 3.23 | 5.92 ± 1.07 | 600.33 ± 123.75 |
| | Week 2 | 5,666.7 ± 344.48 | 104.5 ± 8.22 | 5.00 ± 1.14 | 528.75 ± 149.55 |

(*Cohen, 2021*), knowing that the ES can be classified as small ($0.2 \leq d < 0.5$), medium ($0.5 \leq d < 0.8$) and large ($d \geq 0.8$). The level of significance was established at $p \leq 0.05$.

## RESULTS

A Student's t-test showed non-significant results for the anthropometric measurements, swimming performance (100-m), RPE and all psychological variables ($p > 0.05$) before the intervention at the pretest.

RPE scores were significantly affected by test time ($F = 14.7$; $p = 0.001$; $d = 0.70$), group ($F = 12.2$; $p = 0.002$; $d = 1.02$) and interaction ($F = 25.3$; $p < 0.001$; $d = 0.97$) effects. *Post hoc* testing showed that RPE score was significantly increased in the HIT group ($p < 0.01$) during the intervention period, while the RPE score decreased significantly in the control group ($p = 0.015$, Fig. 2).

A significant effect of time was found for all psychological variables and swimming performance ($F \geq 17.6$; $p < 0.001$; $d \geq 0.97$). Furthermore, a significant group × time interaction effect was found in cognitive ($F = 14.9$; $p < 0.001$; $d = 0.62$) and somatic anxiety ($F = 5.37$; $p = 0.031$; $d = 0.55$) were found. Only a significant group effect was found in somatic anxiety ($F = 27.1$; $p < 0.001$; $d = 1.2$). *Post hoc* comparison revealed that both groups increased their swimming performance significantly with a medium effect. Cognitive and somatic anxiety increased significantly from pre to post test in the HIT group with a large effect, while it did not change significantly in the control group with only a medium effect (Table 2). Self-confidence decreased significantly in the HIT group, while it did not reach significance in the control group, but both had a large effect (Table 2).

## DISCUSSION

This present study aimed to investigate the effect of intensive training on swimming performance and psychological state in competitive swimmers. Our results showed that a 2-week of intensive swimming training block significantly increased state anxiety more than standard training, while swimming performance changed similarly in both groups.

The sudden increase in training intensity had increased the rate of perception effort (RPE, +28.9%), which had a negative impact on the psychological state of the swimmers in the experimental group by increasing the cognitive anxiety (26.8%) and somatic anxiety (15.4%). More specifically, higher intensities than usual could be interpreted by athletes as a sign of fatigue (*i.e.*, mental fatigue, physical fatigue) and as additional pressure, faced with a new level of performance required, which can trigger an anxiety reaction (*Ekkekakis, Hall*
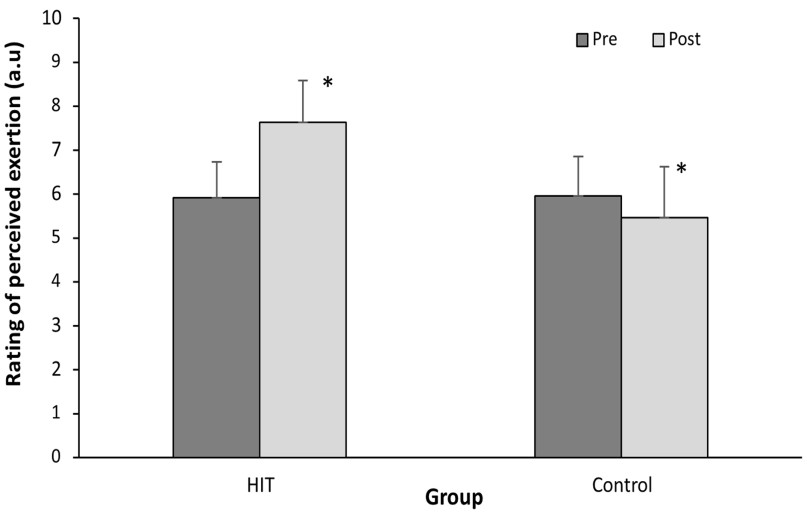

**Figure 2** Change in RPE score between the last 2 weeks before IP and the 2 weeks of intervention period in the both groups. (*) $p < 0.05$.

**Table 2 Changes in psychological state and swimming performance after 2 weeks of training in the both groups.**

| Variables | Groups | Pretest | Posttest | p-value | Effect [95% CI] | Delta change (%) | Cohen's d |
|---|---|---|---|---|---|---|---|
| **T 100 m (s)** | Control group | 57.03 ± 1.12 | 56.47 ± 1.11 | 0.002 | 0.55 [−0.44 to 1.55] | 0.96 | 0.48 [medium] |
| | HIT group | 56.85 ± 1.12 | 56.13 ± 1.24 | <0.001 | 0.73 [−0.33 to 1.78] | 1.28 | 0.63 [medium] |
| **Cognitive anxiety** | Control group | 11.36 ± 1.50 | 12.18 ± 1.08 | 0.167 | −0.82 [−1.98 to 0.35] | 7.22 | 0.55 [medium] |
| | HIT group | 11.18 ± 1.99 | 14.18 ± 1.17 | <0.001 | −3 [−4.46 to −1.55] | 26.83 | 1.91 [large] |
| **Somatic anxiety** | Control group | 14.55 ± 1.04 | 15.18 ± 1.25 | 0.480 | −0.64 [−1.66 to 0.39] | 4.40 | 0.55 [medium] |
| | HIT group | 15.36 ± 1.21 | 17.73 ± 1.10 | <0.001 | −2.36 [−3.39 to −1.34] | 15.37 | 2.15 [large] |
| **Self-Confidence** | Control group | 14.27 ± 1.42 | 13.27 ± 1.19 | 0.063 | 1 [−0.17 to 2.17] | 7.01 | 0.83 [large] |
| | HIT group | 14.18 ± 1.33 | 12.91 ± 0.83 | 0.020 | 1.27 [0.29 to 2.26] | 8.96 | 1.05 [large] |

*& Petruzzello, 2005*). Anxiety can also lead to an increase in RPE, creating a negative feedback cycle where anxiety increases RPE, and vice versa. In particular, anxiety can decrease athletes' ability to regulate their emotions, which may make them more sensitive to signals of fatigue and discomfort during training, thus increasing their perception of effort (*Parfitt, Rose & Burgess, 2006*; *Borg, 1998*). On the other hand, a new level of intensity imposed during training can cause physiological maladjustment (*i.e.*, metabolic change, lactic accumulation, muscle fatigue), which could lead to psychological pressure and consequently an increase in the state of anxiety in athletes (*Lehmann et al., 1997*). In return, our results showed an increase in the state of cognitive anxiety (7.22%) after standard training despite the fact that the RPE value was decreased (−8.4%). Other factors could be the causes of these results such as the physical state of the swimmers (*e.g.* muscular exhaustion, insufficient recovery) (*Lehmann et al., 1997*).

Self-confidence was negatively influenced after intensive training (−8.96%) and after standard training (−7.01%). In fact, a clear link between increased anxiety and decreased

self-confidence in elite athletes has been shown in several previous studies (*Mesagno et al., 2019*; *Eysenck et al., 2007*). *Woodman & Hardy (2003)* had shown that high levels of anxiety can lead to negative thoughts and doubts about personal abilities which can reduce their confidence in achieving optimal performance. Furthermore, the self-concept model in sport psychology suggests that anxiety can negatively influence athletes' self-confidence by affecting their perception of themselves as competent athletes, leading to a decreased self-confidence (*Marsh & Martin, 2011*). Additionally, anxiety can disrupt the concentration and mental focus needed to achieve peak performance. This mental distraction can lead to lower self-confidence (*Eysenck et al., 2007*).

Our results revealed that 100-m swimming performance was significantly increased in the HIT group (1.28%) and in the control group (0.96%). These results show the absence of negative effects of two training conditions on swimming performance. In fact, the high level of training experience could be an important factor that helped reduce the negative impacts of increased anxiety and decreased self-confidence on swimming performance in competitive swimmers who participated in this study. More specifically, experienced athletes tend to develop more effective strategies for managing training-related anxiety (*Hanton & Jones, 1999*). Their experience allows them to recognize and control symptoms of anxiety, such as increased heart rate or negative thoughts, allowing them to maintain focus and performance during training. In addition, experienced swimmers tend to have a broader repertoire of coping skills to deal with anxiety. This may include relaxation techniques, specific pre-competition routines, visualization strategies that help them stay calm and focused during training (*Mellalieu, Hanton & Fletcher, 2006*).

To summarize, a sudden increase in training intensity over 2 weeks increased state anxiety more than standard training, with a similar improvement in swimming performance. It may be that the negative levels of psychological variable results in this study did not exceed the limit that could affect swimmers' performance. This negative level of psychological variables is not of concern so far, but it must be constantly monitored by psychologists and psychological preparation programs must be included in order to improve the mental and psychological abilities of competitive swimmers. In addition, it is recommended to gradually increase training intensity to reduce the risk of muscle injuries and mental fatigue, which could positively promote the psychological state of competitive swimmers.

This study has certain limitations that must be addressed as well. For instance, the absence of an electronic time recording system regarding the swimming performance times. The use of other variables (*e.g.* perception of fatigue, stress, dietary index measurements) could provide us more insights into the effect of HIT on the psychophysiological state in competitive swimmers. Additionally, future research should include female swimmers to learn more about the effect of HIT on gender-specific psychophysiological responses. Lastly, future research should investigate the effect of HIT influences physical and mental status at the level of other age categories (*i.e.*, prepubescent, master swimmers).

## CONCLUSIONS

Our findings revealed that the sudden increase in training intensity increased state anxiety more than standard training, while similar improvements in swimming performance were observed after both training conditions. Although the negative level of state anxiety and decreased self-confidence in this study are not of concern so far, it is advisable to monitor it continuously as it over a longer training period perhaps could have a negative influence on swimming performance. In addition, it is recommended to strengthen the psychological state of swimmers by including mental preparation programs in training strategies in order to reduce the risks of mental and physical problems that could appear during these types of training.

## ACKNOWLEDGEMENTS

The authors thank all the subjects who participated in this study.

### Funding

The authors received no funding for this work.

### Competing Interests

Tiago M. Barbosa is an Academic Editor for PeerJ.

### Author Contributions

- Hajer Aouani conceived and designed the experiments, performed the experiments, analyzed the data, prepared figures and/or tables, authored or reviewed drafts of the article, and approved the final draft.
- Sofiene Amara conceived and designed the experiments, performed the experiments, analyzed the data, prepared figures and/or tables, authored or reviewed drafts of the article, and approved the final draft.
- Faten Sahli conceived and designed the experiments, performed the experiments, analyzed the data, authored or reviewed drafts of the article, and approved the final draft.
- Tiago M. Barbosa conceived and designed the experiments, performed the experiments, analyzed the data, authored or reviewed drafts of the article, and approved the final draft.
- Nizar Souissi conceived and designed the experiments, performed the experiments, analyzed the data, authored or reviewed drafts of the article, and approved the final draft.
- Roland van den Tillaar conceived and designed the experiments, performed the experiments, analyzed the data, authored or reviewed drafts of the article, and approved the final draft.

### Human Ethics

The following information was supplied relating to ethical approvals (*i.e.*, approving body and any reference numbers):

The Research Unit (UR17JS01) Sports Performance, Health and Society, Higher Institute of Sport and Physical Education of Ksar Saîd, University of la Manouba, Tunis, 2010, Tunisia. (protocol code 2022-167).

## Data Availability

The raw data is available in the Supplemental File.

## Supplemental Information

Supplemental information for this article can be found online at http://dx.doi.org/10.7717/peerj.17708#supplemental-information.

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
