# Peer review of "Effect of intensity training block on anxiety state and performance in competitive swimmers"

_PeerJ, doi:10.7717/peerj.17708_

## Round 0.1 · original submission · Major Revisions

Dear Authors,

Congratulations on your work. Reviewers have raised several concerns about the manuscript. I would highlight the need to improve your rationale in the introduction and provide a more detailed description of your methods and the statistical analysis. Also, you need to review your paper for language and typos. Adjustments in your figures are also recommended, mainly for including statistical elements (i.e. standard deviation bars). Your results section is confusing, and some tests described in the analysis section are missing (i.e. two way ANOVA).

Please consider each of the reviewers' comments below.

·

Basic reporting

The authors examined the effect of a 2-week block periodization scheme vs. standard aerobic
training on anxiety state and 100m swimming performance in male competitive swimmers. I believe no serious grammar and spelling issues need to be solved. The Figures and Tables provide an effective visual presentation of the research findings.

Experimental design

The Introduction part is successfully identifying the previously unexplored area of research, as the topic discussed is generally underrepresented in the existing literature. However, the authors failed to lead toward the study hypothesis, as more background information should be included in lines 92-95. More information can be derived from the study by Clemenetes et al. (2021), already included in the references list, or the study by González-Ravé et al., 2021 (DOI: 10.1123/ijspp.2020-0906). The Methods are described in detail and the statistical analysis is sufficient.

Validity of the findings

Lines 36-38 are somewhat confusing: Do these results concern the EXP group? Please re-write this part. Line 265, I believe something is missing in this paragraph.
A major flaw I see here, and not mentioned in the Limitations paragraph, is the lack of an electronic time recording system regarding the 100-m maximum performance times.
Where the authors used the mean value (of the two timekeepers) for further analysis?
In addition, why swimmers used a block start during the max time trial? Wouldn't be better to use an in-water start to minimize the effect of an enhanced start between the pre - and post-measurement?
In the Discussion part, I think that the authors should try more to justify the lack of significant improvement during the 100m performance. For example, you are mentioning that Costill et al. attributed the nonexistence of swimming performance improvement to muscle fatigue and the inability to ingest enough carbohydrates.
Can you also give some speculations, since you did not measure BLa, CK, or dietary indices? In addition, why do the authors suggest "gradually increase the intensity of training to avoid the risks of overtraining" since no such evidence is derived from the results (existence of muscle fatigue in the participants)? Please double-check the references part, there are many mistakes and inconsistencies, mostly in how the Journals are presented (e.g., capital letters, abbreviations, italics, etc.)

Reviewer 2 ·

Basic reporting

The authors examined the effect of an intensive training block (HIT) for two weeks on the anxiety state and swimming performance compared to standard aerobic training. The main results of the study demonstrated that a sudden increase in training intensity negatively affect the psychological state of competitive swimmers and, additionally, no significant change was observed sprint performance after two weeks of training,

Major comments and suggestions.
The manuscript is well organized, but the rationale and gap in the literature is not clear. Some references utilized for rationale to describe the HIT are very old. There are robust evidence body in the topic. I suggest you see paper of Buchheit and Laursen, 2013 in order to use more recent terms related to the studied topic. Moreover, design is not appropriately described in the manuscript as well as some methods should be better described for clarity.

Specific comments and suggestions.
I understood that the central issue of the manuscript refers to effects of HIT on anxiety. The authors refer that studies are required for clarification of psychological trajectories (e.g. cognitive anxiety, somatic anxiety and self-confidence), but is necessary to identify some rationale for it. Just absence of studies is not so good justification. Moreover, you showed Dalamitros et al., 2019 did not observed psychological changes after lower limb strength exercise and, then, based on this idea, it is necessary to know if two weeks of high intensity training (intensity > 80% of Heart rate max) are able to influence psychological variables and swimming performance in competitive swimmers. There is no relationship between the studied and your rationale. Please, improve the rationale! Why can HIT influence anxiety in swimmer?

It is necessary to describe what is cognitive anxiety, somatic anxiety and self-confidence. These concepts are presented in introduction sections but is not appropriately described.

Experimental design

Specific comments and suggestions (materials and methods).

Study design
It not appropriately describes objective in this section. Please remove.

It is necessary to describe experimental design (crossover, RCT?).

Participants
Sample size estimation should be describer in the statistical section.

Why sample size estimation was based on an independent t-test, and then you analyzed the data using ANOVA?

You considered and d = 1.29 for sample size estimation. This score is quite big, and you should describe why you choose this score.


Swimming training
Why 2 weeks of HIT training?

Statistical section
This section is hard to be interpreted due to the imprecise description of the experimental design. Moreover, the inferential analysis is not coherent to sample size estimation.

Validity of the findings

Specific comments and suggestions (Results).

In the results, you show in Figure 2 RPE. As there are no indications of statistically significant differences, I am assuming that the groups were comparable. It is critical for you experiment because the effects of HIT in anxiety should be dependent on HIT effects. Based on this, the anxiety difference observed between HIT and Standard should be attributed to what?
It is critical for you discussion and should be revised with caution!

·

Basic reporting

General comments:

The study is interesting, methods are adequate, I think those perceptive/psychological variables are very interesting to be investigated under the study conditions and the results provide interesting information for both scientists and coaches.
However, English sounds a bit “naïve”, I suggest writing to be reviewed by a fluent English speaker, even though no major mistakes are present.

Abstract:

Line 27: I would avoid the term “very relevant” and choose a more conservative expression. Suggestion: the emphasis on the relevance of the investigation would be kept if the authors just say “relevant”
Line 40: “In summary” here sounds redundant.

Introduction:

Lines 54-55: “Maintenance” instead of “maintains”. Content between parenthesis (….) should be incorporated into the sentence for better fluency.
Line 57: what is meant by “psychological trajectories”?
Line 59: try to avoid the term “very” here.
Line 62: short distances (<20 seconds). I would rather use “short duration”.
Line 81: I have the impression that the “effect of coaching” is not what is intended to be said: “Effect of training instead”?
Line 98: increases

General comments:
-even though increased training loads may lead to adaptations disturbances (when in excess when insufficient recovery is provided etc), positive adaptations can only be achieved through well-planned increases in training load. I would be more careful in expecting that training load increases per se may cause psychological disturbances, as your hypothesis states.
- I would exclude the performance maintenance (non-improvement) from the hypothesis since effects on performance are not the main objective of your study and are not expected anyway due to the short study duration.

Experimental design

Methods:

Line 104: microcycle
Line 106: Please observe that you here use “pre-tests” and “post-tests” but on the tables you use “pretest” and “posttest”. You have to choose one way and keep it all over the text.
Line 176: data sets.
General comments:
- Please inform that the participants were all men (male subjects).
- Variables are adequate and interesting, and the main study question is relevant.
- Methods seem in general adequate.

Validity of the findings

Results:

Line 187: Student´s t-test
Line 195: group effect
Lines 195-196: I suggest you avoid the term “only” here. It gives the impression of a minor importance.
Lines 205-206: “both groups”, “was observed”. Check the writing of “pretest”.



Discussion:

Line 215: “Aquatic training bloc” seems inadequate here. “Intensive swimming training block” is my suggestion. I understand that you use “aquatic” for a variation in writing style, but I would not recommend it here.
Lines 219-221: here you state that the increased RPE negatively influenced the state of anxiety in the HIT group. Even though this is very well possible, I would be a bit more conservative in stating this in such a causal relationship. It is interesting enough that both phenomena were observed (increased RPE and altered anxiety). If there is a possible causal effect, this must be pointed out as a possibility. Especially if literature has already observed it and suggested a causal relationship (Selmi 2018, Barroso 2015).
If there is a logical background supporting an eventual causal relationship, I would suggest that you explore this.
Line 248: what is meant by “state of pressure”? I can maybe understand what you mean but you should find another expression. If this is an expression used in Costill´s work, please let it be noticeable.
Line 257: not “the presence” but “the use” of other variables…
There is a missing part in your last sentence.

Conclusion:

Line 265-266: here again there is a causal relationship. I don´t think you should go so far. Increased training intensity led to alterations in both anxiety and self-confidence. This is interesting enough and a piece of important information provided by the study. Not necessarily one cause the other.

---

## Round 0.2 · Minor Revisions

Dear authors,

Please, provide further responses regarding the following issues:

- Please, review your terminology. HIT is different of HIIT, in my opinion, your work is about the second.
- L.154. The authors stated that “The HIT group was invited to perform an intensive low-volume training (total distance = 155 km): 60% aerobic training, 40% HIT”. Please, review your terminology since HIIT is also predominantly aerobic training.
- L154. The authors stated that both groups perform the same training protocols. Thus, what changed between groups was the training volume/load, since the standard group also performed HIIT (10%). Please consider reviewing your narrative to be precise and consistent regarding the topic being investigated. For example, the term “intensive training block” in the abstract seems more appropriate than the “high-intensity training” used in the title.
- Please, consider using Shapiro-wilk’s test instead Kolmogorov, it is more appropriate to small samples.
- I was not able to find the ICC results descried in the Methods.
- L.216. There are some typos in this paragraph.
- L.245. Please, review the final sentence of this paragraph. Im not sure if the idea presented I clear.
- Conclusion: please remove the causal idea between raised effort perception and the state of anxiety and self-confidence. It’s is clear that those variables were correlated but I have doubts about causation. Your own text speculate that effects could rely in other causes.
- Regarding sample size estimation, the software you used has the ANOVA option, it is under the F tests category. Also, the print screen you provided in your response suggests your ES estimation was based on your age data. Is this correct?
- The authors stated that coaches should progressively increase intensity, however, it was not clear how harmful was the effect found. Could be those changes in a health range? Should be those changes actually expected as a markers of raising training load? Is that harmful if made for such short periods?

Please address those open issues in your result interpretation.

·

Basic reporting

The revised manuscript is now well-written and effectively organized, providing a comprehensive review of relevant literature. The language is concise and accessible.

Experimental design

A well-defined research question is now apparent. The methodological approach is well-described and enhances the credibility of the study's findings. The authors utilize appropriate research design, data collection methods, and statistical analyses.

Validity of the findings

The study presents novel findings that advance understanding within the field.

Additional comments

I have no other comments. The authors successfully addressed all my concerns.

---

## Round 0.3 · Minor Revisions

Dear authors,
Thank you for considering some of the concerns raised by me and the reviewers. However, I believe that a more careful revision is needed.
For example...
- The terminology of HIIT / HIT was questioned, which means that the definition was not clear or precise, but the authors does not made any change. ALso, the justification to use HIT for being "anaerobic" is contradictory to the characteristic of the training protocol applied.
- Another question about HIIT terminology made the authors change the title, however, the response letter says "This present study is about the high intensity training".
- The sample size calculation was also questioned and the response suggested that authors used the age data to run it. However, age is not the outcome.
- The authors were questioned about practical applications and the magnitude effects of their findings in a clinical view. How harmful were the findings? However, in the response letter the authors stated about "to avoid the risks of overtraining (excessive increase in anxiety)". The question remains, the findings were that harmful? No inclusion in text were found about that.
- I would add that is not clear why the 100m results are presented twice and by different values (mean and median).
- Also, please review your results section for clarity. I'm not sure what (0.001< p ≤ 0.048) means.
- Please, report the ANOVA interactions and consider it before the pairwise comparisons.
Thank you for consider these remaining concerns.

---

## Round 0.4 · Minor Revisions

Please, review for terminology consistency throughout the paper. For example, “bloc” vs “block”.

Please, inform the origin of data used in the new sample size calculation.

Please, review your results section for clarity and precision.

For example: “A significant effect group was observed only at the level of cognitive and somatic anxiety (p < 0.05; 0.64 ≤ ES ≤ 1.53)”. In this sentence, the ES was based on eta²? The 0.64 refers to cognitive or somatic? Also, provide the exact p-value. I suggest to use the following structure:
A significant effect group was observed only at the level of cognitive (F= X.XX; Eta² = X.XX; p = X.XX) and somatic anxiety (F= X.XX; Eta² = X.XX; p = X.XX).

The same for the following sentence presenting specific values for each psychological variables, including interactions. “Moreover, a significant effect of time was found only in the psychological variables (p < 0.01; 0.98 ≤ ES ≤ 1.37). A significant effect of group × time interaction was showed only in the psychological variables (p < 0.05; 0.55 ≤ ES ≤ 0.79).”

Please allow me to reinforce the relevance of this description. I tried to replicate the authors' analysis and I found some inconsistencies. I’m not saying that the analysis was wrong, but the description does not allow precise replication. For example, I was not able to find significant between-group effects for cognitive anxiety (p= 0.12). Also, I do not found a significant interaction for self-confidence (F= 0.25; eta2= 0.13; p= 0.62). However, I found a significant time effect for 100m performance (p<0.001).

Regarding RPE data, the author stated that: “Our results showed that RPE score was significantly increased in HIT group (p < 0.01; + 28.9%) during the intervention period. However, the RPE score remains unchanged in the control group (p > 0.05) (figure 2)”. Were those p-values from ANOVA or t-tests? Please include SD data in the figure and remove the % change. Also, the control group RPE data do not show a normal distribution, please consider the non-parametric analysis and data presentation.

Once again I ask the authors to provide a further interpretation regarding the magnitude of their findings. Considering the absence of group differences in psychological variables and the absence of negative effects on performance, how harmful are the risks of this sudden intensity increase? Were the values found alarming? My point is, those psychological effects may be expected along training phases, while tapering or gradual increases are not always possible depending on the competitive schedule. Thus, I ask the authors to include in the text evidence supporting the need for a gradual intensity increase based on the range of the values found.

---

## Round 0.5 · Minor Revisions

Dear authors, thank you for addressing the points raised during the reviewing process. Most relevant issues were successfully changed and, the manuscript is close to being suitable for publication.

Before going further, please review your conclusions in the abstract and the main text. This section is too long and most of it (L306-311) is identical to a paragraph already presented in the discussion (L285-290). I suggest being concise and restricted to your findings.

---

## Round 0.6 · accepted · Accept

Dear authors, thank you for addressing all the issues raised. I have assessed the revision and I'm happy with the current version. In my opinion, the manuscript can be accepted for publication.

Regards